# Early life environment moderates association of body composition and internalizing problems in adolescence
Claudia Buss [1,2,3], Alice M. Graham[4], Lauren E. Gyllenhammer[1,2], Pathik D. Wadhwa[1,2,5,6,7] & Jerod M. Rasmussen [1,2,7,8] ✉

Metabolic and depressive disorders are major chronic global health concerns, often co-occurring and mutually reinforcing each other. Thus, understanding risk and protective factors underlying their development is crucial for identifying effective preventive strategies. Participants included N = 10,446 participants (31,418 observations) from the Adolescent Brain Cognitive Development Study aged 10-15 years. Primary outcomes were internalizing problem scores, and random slopes quantifying the within-person coupling between waist-to-height ratio and internalizing problems. Predictors included early-life adversity measures and potentially protective environments measured at the family, community, peer, and school level. Early-life adversity and protective environment scores were examined as moderators of the coupling between body composition and internalizing problems. Early-life adversity was significantly associated with the magnitude of within-person coupling (random slope); individuals with higher early-life adversity exhibited a stronger coupling between waist-to-height ratio and internalizing problems ($r^2$=4.6%, t = 26.6, $p < 10^{-10}$). The adversity-related amplification of waist-to-height ratio and internalizing coupling was mitigated by the protective environment score (t = -5.3, $p < 10^{-6}$), with family and community components showing the strongest effects. Early-life adversity intensifies the coupling between waist-to-height ratio and internalizing problems, but protective environments may mitigate these effects. These findings motivate research into interventions that reduce early adversity and strengthen protective environments to improve youth mental and physical health.

Metabolic and depressive disorders are pervasive global health issues. Early risk factors—such as elevated body-mass-index (BMI) and waist-to-height ratio (WHtR) for metabolic disorders, and internalizing symptoms (e.g., depression, anxiety, withdrawal) for mental health problems—often emerge in childhood and adolescence[1]. Adolescence is a key period during which risk factors become more pronounced, shaping future disorder susceptibility[2].

While physical and mental health are widely studied, research—and clinical practice—often examines them in isolation. However, their tendency to co-occur and reinforce each other[3–6] is well-established. For example, one recent meta-analysis estimated that individuals with

metabolic syndrome have a 49% higher likelihood of developing depression (a pattern that may reflect a broader vulnerability to internalizing symptoms), while those with depression have a 52% increased risk of developing metabolic syndrome[7]. Given the public health burden of metabolic and depressive disorders, and their bidirectional, self-reinforcing relationship, a deeper understanding of how this link emerges is urgently needed. Further, physical and mental health outcomes are increasingly recognized as being shaped by early-life conditions. However, the early-life factors that contribute to the emergence of this bidirectional, self-reinforcing relationship during adolescence remain poorly understood.

[1]Development, Health and Disease Research Program, University of California, Irvine, CA, USA. [2]Department of Pediatrics, University of California, Irvine, CA, USA. [3]Charité – Universitätsmedizin Berlin, corporate member of Freie Universität Berlin and Humboldt-Universität zu Berlin, Department of Medical Psychology, Berlin, Germany. [4]Department of Behavioral Neuroscience, Oregon Health & Science University, Portland, OR, USA. [5]Department of Psychiatry and Human Behavior, University of California, Irvine, CA, USA. [6]Department of Obstetrics & Gynecology, University of California, Irvine, CA, USA. [7]Department of Epidemiology, University of California, Irvine, CA, USA. [8]Department of Biomedical Engineering, University of California, Irvine, CA, USA. ✉e-mail: rasmussj@hs.uci.edu

Clinical, psychosocial, and biological evidence support a bidirectional relationship between metabolic and depressive disorders. For example, excess weight can lead to negative self-esteem and internalizing symptoms such as withdrawal and anxiety, which are then further exacerbated by stigma and peer rejection due to difficulties in participating in physical activities[8]. Biologically, *central* adiposity is linked to inflammatory processes[9] that are well-established to have effects on mood[10]. Conversely, emotional eating is a common feature of depression in youth, while comfort feeding is a well-known parental behavior aimed at reducing stress and anxiety in children[11]. Thus, given the strong evidence for mutual influence between physical and mental disorders, this relationship likely gets stronger over time, leading to further decline in physical and mental health.

While well-documented in adults, the onset of their emergence remains less understood. For example, body composition proxies (i.e., BMI) and internalizing problems have been linked to each other in a large longitudinal early childhood (~1.5–6 years old) sample[12]. More recently, in a large representative sample of adolescents across the US, researchers demonstrated associations between baseline internalizing scores and BMI change (10–12 years old), and baseline BMI with depressive symptoms change[3]. Here, in the same cohort, we aim to build upon these findings by: (a) extending the timeline of observations (up to 15 years of age), (b) considering a potentially more informative measure of adiposity[13] (WHtR instead of commonly used BMI), and (c) explicitly examining early-life influences on the *coupling* between physical and mental health measures.

Identifying common risk factors that drive the self-reinforcing cycle of physical and mental health problems—along with protective factors that promote resilience—is essential for creating effective prevention policies and practices. Early-life adversity (e.g., adverse childhood experiences [ACEs]) is a well-established risk factor for adverse physical and mental health outcomes, including obesity and internalizing symptoms, that become more pronounced during adolescence[14,15]. Early-life stress has also been associated with biological mediators of these relationships, including dysregulated stress responses and heightened inflammation[16]. In contrast, identifying protective factors that can reduce risk for negative physical and mental health outcomes and their coupling in children exposed to early-life adversity is equally important.

In this study, we operationalize protective factors based on a theoretical framework previously described by the ABCD Culture and Environment Workgroup[17], which identifies four telescoping domains of environmental influence (family, peers, school, and community) as central to shaping adolescent development. Youth who report higher family support tend to show lower levels of internalizing symptoms[18], including depression and anxiety, and adolescents who regularly share family meals have reduced odds of developing overweight or obesity over time[19]. Similarly, greater peer support is associated with lower internalizing symptoms[18], and peer relationships have also been linked to reductions in BMI, as evidenced by recent meta-analytic findings[20]. Within school contexts, stronger school connectedness has been shown to predict lower depressive and anxiety symptoms[21], and students with higher connectedness scores are more likely to meet physical activity recommendations[22]. At the community level, adolescents who feel a stronger sense of neighborhood connection tend to report fewer depressive symptoms[23], and environments that promote community cohesion, such as those with walkable infrastructure and access to green space, are negatively associated with adolescent BMI[24]. Thus, these four domains are well-suited for identifying modifiable protective experiences that may speak to variability in outcomes (i.e., resilience) by buffering the effects of early adversity and inform future research on targeted intervention strategies.

This study uses a large, multi-site adolescent cohort (ages 10–15; $N \sim 11,000$) with up to five observations across four years to examine how body composition relates to internalizing problems, and how early-life conditions influence this relationship. We focus on internalizing problems due to their clinical relevance as indicators of future emotional distress and risk for later mental health disorders, including but not limited to depression, and their established associations with BMI. Based on the logic that

central adiposity is more directly related to the metabolic and immune consequences of obesity than total adiposity or weight-based measures of body composition (i.e., BMI) alone[9], we test the hypothesis that WHtR (a normalized measure of central adiposity) is a stronger predictor of adolescent mental health problems than BMI. Because central adiposity is thought to be more directly related to metabolic and immune consequences of obesity than total adiposity[13], we extend our analysis using the longitudinal design to test whether the *within*-person relationship between WHtR and internalizing problems is moderated by prior early-life adversity exposure. Here, early-life adversity exposure is operationalized using a previously described proxy[25] for traditional ACEs[26] measures based on a comprehensive set of ABCD study data (ABCD-ACEs, see "Supplementary Materials S1" for further details). Finally, we explore protective factors in this framework. We do so by testing the hypothesis that protective factors, in general, buffer the association between ABCD-ACEs and the *within*-person relationship between body composition and internalizing problems. We then conduct follow-up analyses at the domain- and item-levels aimed at mitigating the negative consequences of early-life adversity.

## Methods
### Sample description
This study used data from the Adolescent Brain Cognitive Development^SM (ABCD) Study (https://nda.nih.gov/study.html?id=2313; https://doi.org/10.15154/cqdy-5453)[27]. ABCD is a longitudinal study of ~11,000 participants aged ~10–15 years of age recruited at 21 sites across the United States, designed to capture a representative sample of sociodemographic variation from 2017 to present. Five study visits were available (baseline through 4-year annual follow-up, see Fig. 1b for age distribution) at the time of analysis. Ethical review for the ABCD study data collection has been described elsewhere. This study was not preregistered.

Data were sourced from ABCD release 5.1 ($N_{\text{baseline}}$ = 11,868). The following variables were considered: predictors (BMI, WHtR), outcomes (internalizing problems), effect modifiers (ABCD-ACEs, protective environments), and covariates (age, sex, pubertal status, area deprivation index, parental education, household income, child ethnicity, and race). Sex was based on caregiver-reported sex at birth. After data cleaning, $n$ = 10,446 participants with a total of 31,418 observations remained (Fig. 2; see "Supplementary Materials S1" for a detailed accounting of data cleaning procedures). Secondary analyses on deidentified data are considered non-human subjects research, and this designation was filed with and approved by the Internal Review Board at the University of California, Irvine. Ethical approval of ABCD study protocols was given by the central Institutional Review Board, University of California, San Diego and individual study sites relied on local IRB approval. Caregivers provided written informed consent, and children provided written assent[28].

### Data analysis overview
The analyses include three complementary parts. First, we compare BMI and WHtR to determine which better represents the relationship with internalizing problems, providing clinical insight. Second, we quantify the effect of prior exposure to early-life adversity on the magnitude of the within-individual association between body composition and internalizing problems. Third, we examine whether protective environmental influences can mitigate the effects of early-life adversity on the relationship between body composition and internalizing problems. All reported statistical tests were two-sided.

### Data analysis: body composition association with internalizing problems
The association between body composition and internalizing problems was tested using a generalized additive mixed model (GAMM, R package *gamm4*). To test whether WHtR is a more clinically useful measure than BMI, we optimized models for predictive quality and complexity. First, to determine which variables are best modeled by either linear or non-linear terms, we considered a full model with all measures constructed as first-

**Fig. 1 | Adolescent body-mass-index (BMI), waist-to-height ratio (WHtR), and internalizing behaviors in the ABCD study. A** Conceptual model comparing the strength of association between BMI (a measure of relative global mass) and WHtR (relative central adiposity/shape) with internalizing behaviors based on the child behavioral checklist (CBCL). **B** Five-yearly study visits were included for the study and centered on ages 10–14, respectively. **C** WHtR was associated with internalizing behaviors across childhood/adolescence in a longitudinal mixed-effects model. **D** WHtR discordance between sibling pairs (a pseudo-control for environmental and genetic influences) was associated with internalizing behaviors discordance at baseline, 2-year, and 4-year follow-up visits (not significant at 1-year and 3-year visits). $n_{base}$ = 1403; $n_{2y}$ = 1098; $n_{4y}$ = 355 sibling pairs.

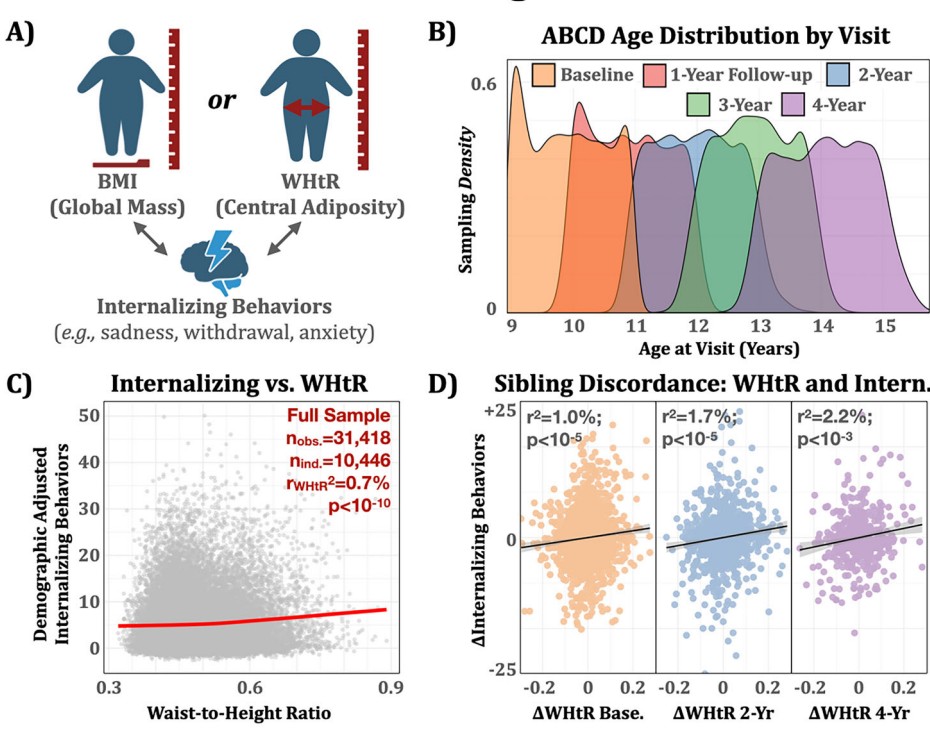

## Adolescent Body-Mass-Index, Waist-to-Height Ratio, and Internalizing Behaviors

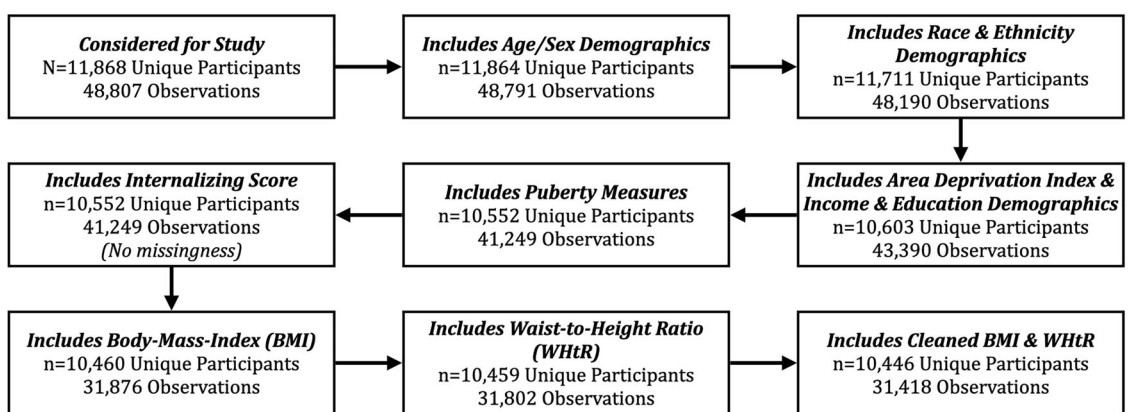

## Data Inclusion Flow Diagram

**Fig. 2 | Data inclusion flow diagram.** Data available for initial analyses are depicted above with respective filters for inclusion/exclusion. Of the final $n$ = 10,446 participants available for study, the following had 1/2/3/4/5 repeated measures available: $n$ = 778/2213/4260/2541/654. For moderation, all individuals had measures of Adverse Childhood Experience measures (ABCD-ACEs) available, whereas $n$ = 7131 (68%) had a protective environment summary score available for analysis.

order (linear) and spline-based higher-order terms. Having identified that *only* BMI has a non-linear form in the context of internalizing problems, we specified all covariates as linear effects and BMI and WHtR (for effect size comparison with BMI) as non-linear effects in GAM models. Next, we compared models with BMI, WHtR, or both as correlates of internalizing problems. Decisions for parsimony were based on additional explained variance ($R^2$) and Bayesian information criterion (BIC), and tested using nested models in an ANOVA framework (see "Supplementary Materials S2: Mixed Effects Generalized Additive Model Selection" for full models and details). In addition, cross-sectional analysis was conducted to provide age-specific effect sizes. All models included random intercepts for site, participant ID (longitudinal models only), and family structure (i.e., siblings) to

account for clustering effects, ensuring that associations between BMI, WHtR, and internalizing problems were estimated while adjusting for shared variance within individuals and families. Further sensitivity analyses were conducted to establish the robustness of findings to several modeling assumptions: error structure (i.e., Poisson; see "Supplementary Materials S2: Mixed Effects Generalized Additive Model Selection" for details), rater (i.e., self-report and teacher report; see "S3 Sensitivity Analyses: Considering Independent Raters of Internalizing Problems"), and age- and sex-specific associations (see "S4 Testing for Age- and Sex-Specific Associations").

We also employed a sibling discordance analysis to better control for unmeasured genetic and environmental confounds associated with WHtR and internalizing symptoms. This approach examines WHtR and

internalizing differences between siblings that partially control for stable environmental and genetic influences, while adjusting for age, sex, and puberty status. This analysis was conducted cross-sectionally using simple linear regression to simplify the effects of age-related variance and missingness.

## Data analysis: early-life adversity and the association between waist-to-height ratio and internalizing problems

The effect of prior exposure to early-life adversity on the magnitude of the association between body composition and internalizing problems was modeled using within-person analyses. Specifically, we specified a random slope term that provides an individualized estimate of the coupling between internalizing behavior and WHtR across visits. These random slope estimates will be referred to as coupling throughout the remainder of the manuscript and indicate individual differences in how strongly WHtR and internalizing problems are linked relative to population-level effects. The estimates of individualized coupling between WHtR and internalizing problems were then regressed against ABCD-ACEs in a linear regression model.

## Data analysis: protective environments, early-life adversity, and the association between waist-to-height ratio and internalizing problems

An ABCD-ACEs by protective environments interaction with the within-person random slopes considers the possibility that protective influences might dampen the negative consequences of early-life adversity (or vice versa, ABCD-ACEs diminish protective benefits) on the coupling between body composition and internalizing problems. These models were first constructed based on a Factor Analysis summary score intended to be reflective of the overall quality of protective environments (see "Supplementary Materials S6: Protective Environment Summary Score"). Building on findings from the summary score analysis suggesting that a single score is not fully explanatory of such positive environments, we repeated the models to gain more fine-grained insight by examining scores at different social levels[17] (family, peers, school, and community), as well as at the item level (see "Supplementary Materials S7: Item-Level Moderation of the Association Between ABCD-ACEs and Random Slopes").

## Reporting summary

Further information on research design is available in the Nature Portfolio Reporting Summary linked to this article.

## Results

Table 1 describes the characteristics of the sample used in this study.

## WHtR outperforms BMI as a predictor of internalizing problems

Repeated measures analyses showed a significant association between body composition and internalizing symptoms in adolescence (Table 2). Specifically, BMI and WHtR explained an *additional* 0.54% and 0.62% variance, respectively, over and above a model composed of sociodemographic predictors alone (combined predicted variance = 2.26%). In combination, WHtR and BMI explained 0.69% variation in internalizing problems, over and above the basic model. BIC-based analysis favored the model including WHtR, but not BMI, balancing predictive quality and complexity. Specifically, (a) the WHtR model was compared against the parsimonious model ($\chi^2(2) = 102.0$, $p < 10^{-3}$), (b) the BMI model was compared against the parsimonious model ($\chi^2(2) = 98.3$, $p < 10^{-3}$), and (c) the combined BMI + WHtR model was compared against the BMI-only model ($\chi^2(2) = 23.9$, $p < 10^{-3}$). These findings indicate that WHtR improves model fit relative to the null, BMI does not improve upon WHtR (based on BIC), and adding WHtR to BMI significantly improves fit (but not BIC) relative to BMI only. This longitudinal finding is further supported by cross-sectional analysis (Supplementary Fig. S2). Notably, a significant WHtR by Age interaction effect (linear mixed effect interaction model: $t(25,084) = 2.39$, $p = 0.016$, $b = 0.044$, 95% CI = [0.008, 0.080]; nesting interaction term: $\chi^2(\mathrm{d}f = 1) = 5.7$,

**Table 1 | Descriptive statistics of sample characteristics**

| Variable | Full sample (N = 10,446; 31,418 observations) |
|---|---|
| Age, mean (SD) [range], years | 11.3 (1.4) [8.9 15.8] |
| Sex, n (%) | |
| Female | 14,912 (47.5%) |
| Male | 16,506 (52.5%) |
| Race, n (%) | |
| White | 24,476 (77.9%) |
| Black | 4363 (13.9%) |
| Other/Unknown | 2579 (8.2%) |
| Ethnicity, n (%) | |
| Non-Hispanic | 25,293 (80.5%) |
| Hispanic | 6125 (19.5%) |
| Area deprivation index, mean (SD) [range] | 94.3 (21.0) [1.1 125.8] |
| Parental education, n (%) | |
| ≤High school graduate | 1170 (3.7%) |
| High school graduate | 1765 (5.6%) |
| GED or equivalent | 743 (2.4%) |
| Some college | 3859 (12.3%) |
| Associates degree occupational | 2359 (7.5%) |
| Associate's degree academic | 1802 (5.7%) |
| Master's Degree | 8421 (26.8%) |
| M.D. or Equivalent | 7880 (25.1%) |
| Ph.D. or Equivalent | 1635 (5.2%) |
| Household income, mean (SD) [range] | |
| <$5000 | 662 (2.1%) |
| $5000–$11,999 | 852 (2.7%) |
| $12,000–$15,999 | 621 (2%) |
| $16,000–$24,999 | 1178 (3.7%) |
| $25,000–$34,999 | 1610 (5.1%) |
| $35,000–$49,999 | 2415 (7.7%) |
| $50,000–$74,999 | 4053 (12.9%) |
| $75,000–$99,999 | 4484 (14.3%) |
| $100,000–$199,999 | 10,767 (34.3%) |
| >$200,000 | 4776 (15.2%) |
| Pubertal development scale score, mean (SD) [range] | 2.3 (1.1) [1 5] |
| BMI, mean (SD) [range] | 20.0 (4.8) [10.0 45.0] |
| WHtR, mean (SD) [range] | 0.48 (0.07) [0.32 0.87] |
| Internalizing problems (CBCL), mean (SD) [range] | 5.1 (5.6) [0 51] |
| ABCD-ACEs score, mean (SD) [range] | 2.7 (2.1) [0 19] |
| Positive environment score (SD) [range] | 0.0 (0.6) [−2.4 1.8] |

$p = 0.016$; see also Supplementary Fig. S3) suggested that the association between WHtR and internalizing symptoms is larger with older ages. Finally, while BMI had a non-linear association (roughly U-shaped centered on a BMI ~ 19, Supplementary Fig. S1), WHtR was linearly associated with internalizing problems (Fig. 1c), supporting its use as a practical trans-diagnostic clinical measure by offering a more intuitive, monotonically increasing association.

**Table 2 | Comparative summary of mixed effects generalized additive model: body-mass-index (BMI) and waist-to-height ratio (WHtR)**

| Model prescription | $R^2$ | edf (BMI) | edf (WHtR) | F (BMI) | F (WHtR) | BIC | $\chi^2$ |
|---|---|---|---|---|---|---|---|
| Parsimonious | 2.26% | n.a. | n.a. | n.a. | n.a. | 183,251 | n.a. |
| *Pars. + WHtR* | *2.88%* | *n.a.* | *3.10* | *n.a.* | *58.44\*\*\** | *183,170* | *102.1\*\*\** |
| Pars. + BMI | 2.80% | 5.15 | n.a. | 38.27\*\*\* | n.a. | 183,174 | 98.3\*\*\* |
| Pars. + BMI + WHtR | 2.96% | 5.00 | 1.00 | 8.86\*\*\* | 26.51\*\*\* | 183,170 | 23.9\*\*\* |

Four linear models of internalizing behaviors were compared: a parsimonious model consisting of demographic factors, a model adding in WHtR, a model adding in BMI (but not WHtR), and a model adding in both BMI and WHtR. Body composition measures added over 0.6% explained variance (a ~27% increase). WHtR appeared to explain more variance per degree of freedom than BMI in the full-sample analysis, and the additional explained variance by adding both BMI and WHtR was not substantial relative to WHtR only (no change in BIC). BIC values are from fitted models; $\chi^2$ statistics reflect likelihood ratio tests comparing each model to the nested model immediately above, except for the final row, where WHtR was added to the BMI model (Pars. + BMI). Significance codes: \*$p < 0.05$, \*\*$p < 0.01$, \*\*\*$p < 0.001$.

Subsequent analyses focused on establishing WHtR as a robust correlate of internalizing problems. First, in a sibling discordance analysis, within-sibling differences (discordance) in WHtR and internalizing problems were associated at all visits except for the 3-year follow-up, which had the smallest sample size due to peak COVID disruptions ($t_{base}(1403) = 3.5$, $p_{base} < 10^{-3}$, $b_{base} = 7.7$, 95% $CI_{base} = [3.4, 12.0]$; $t_{1y}(1332) = 2.9$, $p_{1y} = 0.003$, $b_{1y} = 6.1$, 95% $CI_{1y} = [2.0, 10.2]$; $t_{2y}(1,098) = 4.8$, $p_{2y} < 10^{-3}$, $b_{2y} = 11.1$, 95% $CI_{2y} = [6.6, 15.6]$; $t_{3y}(263) = 1.1$, $p_{3y} = 0.29$, $b_{3y} = 5.2$, 95% $CI_{3y} = [-4.3, 14.7]$; $t_{4y}(355) = 3.5$, $p_{4y} < 10^{-3}$, $b_{4y} = 13.4$, 95% $CI_{4y} = [5.9, 20.8]$; Fig. 1d). Effect sizes between siblings were comparable to those unrelated, suggesting a conserved association in the context of relatively controlled environmental and genetic conditions. Second, the association between WHtR and internalizing problems remained significant with comparable effect sizes when considering a multiverse of modeling choices including different raters of internalizing problems (parent, teacher, and self), transformations toward more normally distributed outcomes (e.g., asinh), error models (e.g., Poisson distribution), and sex-specific associations (for a further discussion, see "Supplementary Materials S3 Considering Independent Raters of Internalizing problems", and "S4: Testing for Age- and Sex-Specific Associations").

### Early-life adversity (ACEs) is associated with the within-person coupling between waist-to-height ratio (WHtR) and internalizing problems

ABCD-ACEs had a significant effect on WHtR-internalizing problems coupling ($r^2 = 4.6\%$, or a ~0.1 s.d. increase in slope for each ABCD-ACE reported, Fig. 3). Sensitivity analyses confirmed that this effect is independent of (and strengthened by controlling for) other sociodemographic factors including sex, race, ethnicity, family income, parental education, and area deprivation index ($t_{ACEs}(13,505) = 26.6$, $p < 10^{-10}$, $b_{ACEs} = 0.64$, 95% $CI_{ACEs} = [0.59, 0.69]$).

### Protective environments, early-life adversity, and the association between waist-to-height ratio and internalizing problems

Protective environmental influences appeared to mitigate the negative health effects of early-life adversity. Specifically, a significant interaction was found between ABCD-ACEs and the protective environment summary score on the relationship between WHtR and internalizing problems ($t(7,131) = -6.2$, $p < 10^{-3}$, $b = -0.3$, 95% $CI = [-0.39, -0.20]$, Fig. 4a). Based on the summary score findings, we further considered specific protective environment domains to identify potential intervention targets for future research. Analyses revealed significant moderation (dampening) of the association between ABCD-ACEs and the coupling between WHtR and internalizing problems by community- ($t_{comm}(7125) = -4.5$, $p_{comm} < 10^{-3}$, $b_{comm} = -0.19$, 95% $CI_{comm} = [-0.29\ -0.11]$) and family-based ($t_{fam}(7125) = -3.4$; $p_{fam} = 0.001$, $b_{fam} = -0.06$, 95% $CI_{fam} = [-0.09\ -0.02]$) protective environments in a model including all four domains. There was no statistically significant evidence for an effect of peer or school ($t_{peer}(7125) = -1.2$, $p_{peer} = 0.22$, $b_{peer} = -0.01$, 95% $CI_{peer} = [-0.04\ 0.01]$; $t_{school}(7125) = -0.82$; $p_{school} = 0.41$, $b_{school} = -0.01$, 95% $CI_{school} = [-0.03\ 0.01]$). However, when singled out in

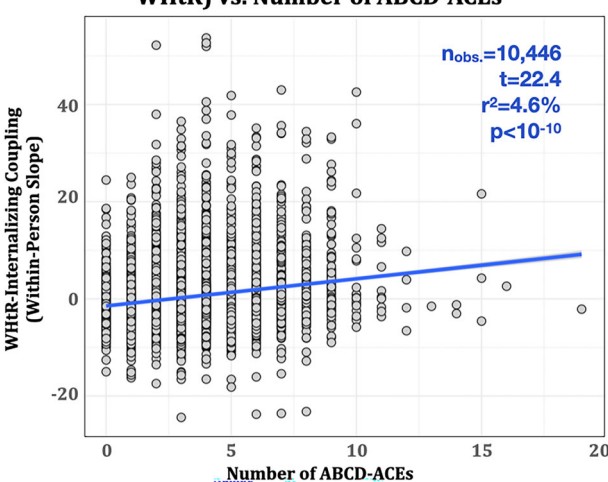

**Within-Person Slope (Internalizing ~ WHtR) vs. Number of ABCD-ACEs**

$n_{obs.} = 10{,}446$
$t = 22.4$
$r^2 = 4.6\%$
$p < 10^{-10}$

**Fig. 3 | The magnitude of coupling between WHtR and internalizing problems is associated with a history of prior exposure to early life adversity.** ABCD-ACEs are associated with the within-person random slope (coupling) between WHtR and internalizing symptoms during adolescence. Note that random slopes represent individual deviations from the population-level fixed effect and are modeled to be centered around zero.

the interaction, peer-based and school-based measures were weakly or marginally significant ($t_{peer}(7131) = -2.5$, $p_{peer} = 0.01$, $b_{peer} = -0.03$, 95% $CI_{peer} = [-0.05\ -0.01]$; $t_{school}(7125) = -1.7$; $p_{school} = 0.08$, $b_{school} = -0.02$, 95% $CI_{school} = [-0.04\ 0.00]$; Fig. 4b). Taken together, this suggests that moderation by peer- and school-based measures is likely confounded by shared variance with community- and family-based domains. Finally, several individual item-level responses in the family domain were identified as moderators of the association between ABCD-ACEs and WHtR-internalizing problems coupling (see "Supplementary Materials S6: Item-Level Moderation of the Association Between ABCD-ACEs and Random Slopes"), suggesting that unconditional family support, belonging, and predictability are especially potent in dampening the negative consequences of early life stress.

## Discussion

We provide evidence from a large, representative longitudinal US cohort, that expands on prior research linking adolescent physical and mental health by: (a) extending the observation timeline (up to 15 years of age), (b) identifying a body composition phenotype (WhtR) tied to internalizing behaviors in adolescence, and (c) characterizing early-life influences on the coupling between physical and mental health measures. Specifically, we demonstrate an association between WHtR and internalizing problems that strengthens across development, consistent with the premise of self-

**Fig. 4 | The magnitude of coupling between WHtR and internalizing problems, particularly among individuals with a history of prior exposure to early life adversity, is smaller among those with a more protective environment. A** A protective environment summary score moderates (dampens) the association between ABCD-ACEs and within-person coupling (random slope) between WHtR and internalizing symptoms in a longitudinal mixed-effects model. **B** Analyses across protective environment domains (family, peers, school, and community) suggest that those at the family level were the most effective at dampening the association between ABCD-ACEs and within-person coupling. *n* = 7125 participants were available for analysis.

## Positive Ecologies Dampen the Effect of ABCD-ACEs on the Coupling Between Waist-to-Height Ratio and Internalizing Symptoms Within Individuals

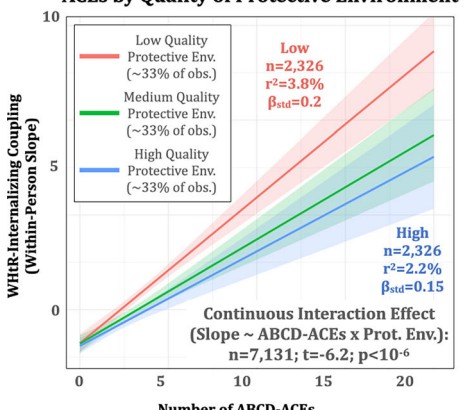

**A) Within-Person Slope (Intern. ~ WHtR) vs. ABCD-ACEs by Quality of Protective Environment**

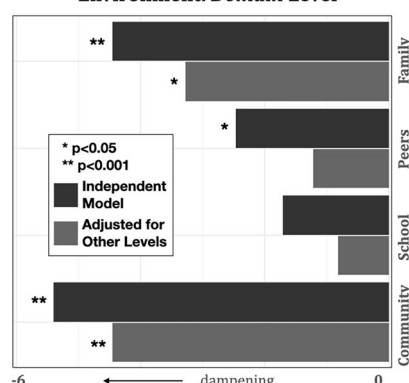

**B) Effect Size of Moderation by Protective Environment: *Domain* Level**

reinforcement and progressive worsening over time, further underscoring adolescence as a potential window for intervention. By comparing siblings in the same family, we provide evidence that the link between WHtR and internalizing problems is not solely explained by genetics or unobserved stable within-family factors. This suggests that differences in siblings' individual life experiences likely play a role in shaping this relationship. Additionally, ABCD-ACEs were associated with within-person coupling between WHtR and internalizing problems, suggesting that early-life adversity increases the strength of the relationship between WHtR and internalizing problems, thus highlighting the role of early adversity in reinforcing cycles of negative health outcomes during adolescence. Finally, we identified key protective environmental factors (e.g., family support) that may mitigate ACE-related impacts on the coupling between WHtR and internalizing problems across adolescent development.

This study adds to evidence supporting body shape indices as preferred phenotypes over BMI for linking body composition with adverse mental health outcomes[29]. Like BMI, WHtR is easily measured in clinic settings and can be widely implemented in routine screenings. However, while BMI demonstrated a U-shaped association with internalizing problems consistent with recent observations[30], waist-to-height ratio showed minimal variation by age or sex in this sample relative to BMI (e.g., $R^2_{BMI,age} = 6.9\%$, $R^2_{WHtR,age} < 0.1\%$; see "Supplementary Materials S8") and demonstrates a linear association with internalizing problems, making it more interpretable for individuals that are underweight, and applicable to smaller datasets. Further, as a marker of central adiposity, WHtR is more likely than BMI to reflect shared biological mechanisms—such as inflammation—linking visceral fat to affective disorders. Collectively, the above considerations make WHtR a practical and needed transdiagnostic measure for understanding the interplay between adolescent physical and mental health outcomes and align with recent American Medical Association recommendations to supplement BMI with other body composition indices[31].

Obesity and depressive disorders are accompanied by common variation in neurobiological, endocrine, immune, and metabolic phenotypes[32] that contribute to symptom progression over time. ACEs have been shown to alter these same biological processes, increasing risk for both obesity and affective disorders and, as suggested by the present study, their coupling. Molecular studies have provided insight into how ACEs become biologically embedded via changes in stress-regulatory systems and epigenetic mechanisms relevant to both metabolic disorders and mental health[33]. However, because the causal direction between these ACE-associated biological and neural circuit alterations implicated in stress regulation remains unclear, more deeply phenotyped longitudinal studies are needed.

Identifying environmental conditions that foster resilience against early adversity is critical to advancing prevention science. Toward this, the current study contributes to the field by elucidating conditions that may buffer the negative impact of early-life adversity on the coupling between body composition and internalizing problems. Specifically, the family-based factors identified as protective highlight potential mechanisms through which acceptance, unconditional support, belonging, and predictability within families may dampen the biological stress response to early-life adversity[34] and, in turn, dampen the coupling between WHtR and internalizing problems. Strengthening these family-based factors, especially in children for whom ACEs originate within the family, as is commonly the case[35], warrants further investigation with individually tailored interventions building on those family-based resources available.

In addition, further research targeting factors at the community and peer-level in interventions is equally important, particularly as community- and school-based factors emerged as significant moderators. Notably, community-based factors demonstrated the largest effects both independently and when accounting for other domains. In contrast, school-based factors did not reach statistical significance. This may reflect limitations in measurement rather than the absence of a meaningful effect, suggesting that more targeted studies with refined school-level assessments are needed to evaluate their true impact.

The effect size of WHtR on internalizing problems pooled across visits ($r^2 = 0.62\%$) is considered small by traditional Cohen heuristics. However, evidence indicates that small studies often overestimate effect sizes, and in the context of the large heterogeneous ABCD study, these effect sizes are relatively substantial[36]. WHtR accounted for a 27% increase in explained variance of internalizing symptoms beyond demographic factors (age, sex, race, ethnicity, puberty status, ADI, parental income, and education), supporting its relative importance. Further, this pooled effect marginalizes across ages, with more pronounced effects seen at later visits (up to 1.7% explained at 14). Similarly, although the reported moderating effect of ABCD-ACEs focuses on the individual-level relationship between WHtR and internalizing problems, this effect likely extends to differences between individuals. This then suggests that the associations between WHtR and internalizing across individuals reported here are likely underestimated for high-risk populations.

### Limitations

This study's strengths, including the large representative sample size and complementary methods supporting robust effects (e.g., sibling discordance, inter-rater comparisons), are balanced by limitations. First,

ABCD lacks deep phenotyping for lab-based measures of central adiposity and internalizing symptoms. Thus, deeper investigator-initiated studies are needed to provide detailed clinical phenotypes and inform on the biological mechanisms of the observed associations. Second, ACE scores were not directly collected in the ABCD study but were rather based on instruments collected with differing sampling strategies, as previously implemented[25]. Thus, this study's operationalization of ACEs did not allow for a deeper investigation of the type of adversity, age at onset, and chronicity that likely uniquely contribute to the coupling between adolescent physical and mental health. While this study focused on adversity and its potential buffering through positive experiences, future work should also examine whether positive experiences in family, school, peer, and community contexts may directly shape the coupling between body composition and internalizing problems. Finally, while this study did not consider brain-based mechanisms due to scope, insights from this work, combined with brain imaging in the ABCD dataset offer abundant opportunities for future research linking brain development to physical and mental health outcomes, including the role of diet[37], adversity, reward, other neurodevelopmental disorders, executive function, and social determinants of health.

## Conclusions

Collectively, this study highlights the importance of WHtR as a meaningful and practical measure of physical-mental health coupling in adolescence, particularly in the context of early-life adversity. Further, it highlights a potential for protective factors, particularly family and community support, that may have benefits to adolescent physical and mental health in the face of adversity. Thus, given the high and increasing prevalence of physical and mental health disorders, policies informed by this study that promote protective environments could yield significant long-term social benefits.

## Data availability

Underlying individual participant data and corresponding data dictionaries are shared as part of the ABCD data repository at the NIH Brain Development Cohorts (NBDC) Data Sharing Platform (https://www.nbdc-datahub.org/abcd-study). All variables used in this paper are denoted by their data dictionary name. Note that future ABCD data release variable names may not be consistent with those used here. All derived data (e.g., random slopes) can be recreated through public access code (https://github.com/jerodras/comms_psych_whtr-int-ela) and will be made available upon request. As with all ABCD data, necessary Data Use Agreements will be required for data access.

## Code availability

All code used to produce the paper's findings will be made available on JMR's GitHub site (https://github.com/jerodras/comms_psych_whtr-int-ela).

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

## Acknowledgements

We thank the participant volunteers for donating their time to the project. In addition, we thank Bianca T. Leonard for useful discussions on figure design. The following funding sources had no involvement in the study: NICHD R00 HD-100593 to J.M.R., R00 HD-097302 to L.G., R01 HD-107176 to P.D.W.; NIMH R01 MH-138481 to J.M.R. The funders had no role in study design, data collection and analysis, decision to publish or preparation of the paper.

## Author contributions

C.B., A.M.G., and J.M.R. contributed to the conceptualization of this project; L.E.G., P.D.W., and J.M.R. contributed to the methodology used; C.B. and J.M.R. have directly accessed and verified the underlying data reported in the paper and contributed to the analysis; and all co-authors contributed to the writing process.

## Competing interests

The authors declare no competing interests.
