## [Transparent Peer Review file · Communications Psychology]

Early life environment moderates association of body composition and internalizing problems in adolescence

Corresponding Author: Dr Jerod Rasmussen

Version 0:

Decision Letter:

Dear Dr Rasmussen,

Thank you for your patience during the peer-review process. Your manuscript titled "Body Composition and Internalizing Problems in Adolescence: Moderation by the Early Life Environment" has now been seen by 2 reviewers, and I include their comments at the end of this message. They find your work of interest but raised some important points. We are interested in the possibility of publishing your study in Communications Psychology, but would like to consider your responses to these concerns and assess a revised manuscript before we make a final decision on publication.

We therefore invite you to revise and resubmit your manuscript, along with a point-by-point response to the reviewers. Please highlight all changes in the manuscript text file.

Editorially, we consider it important that revised manuscript addresses the reviewers' concerns regarding correcting for multiple comparisons and providing measurement models and fit indexes for factor analyses. Please address all methodological and conceptual concerns.

I am attaching an Editorial Requests Table that details critical reporting requirements for the revised manuscript. Please attend to each item and ensure your manuscript is fully compliant. If your revised manuscript is not aligned with these requests on major issues, such as those concerning statistics, it may be returned to you for further revisions without re-review.

Please submit the following items:

- Revised manuscript
- Point-by-point response to the referees' comments
- Cover letter (as a separate document)
- <https://www.nature.com/documents/nr-reporting-summary.pdf>>Nature Research Reporting Summary
- Completed Editorial Request Table (attached).

via this link: Link Redacted .

Additional guidance is available in our style and formatting guide Communications Psychology formatting guide.

Best regards,

Jennifer Bellingtier

Jennifer Bellingtier, PhD
Senior Editor
Communications Psychology

REVIEWER EXPERTISE:

Reviewer #1 adversity, mental health, development

Reviewer #2 adversity, resilience, development

REVIEWER REPORTS:

Reviewer #1 (Remarks to the Author):

This study examined the moderating effects of child risk and protective factors in the association between body composition and internalizing symptoms. Data were drawn from the ABCD Study and findings suggested that higher adversity exposure was associated with stronger coupling between body composition and internalizing symptoms. Certain protective factors (family and community) mitigated this effect. The study benefits from a large, heterogeneous sample and a thorough analytic approach. At the same time, I have several suggestions.

Throughout the manuscript, it would be helpful to have greater specificity in the description and labeling of constructs. For example, in the adversity literature, "ACEs" often refers to a specific list of exposures from the original CDC ACEs study. Because the metric of adversity in the present study is more comprehensive, it may be helpful to instead label the construct as early-life adversity (ELA). Similarly, the introduction is largely focused on depression specifically, but the metric used in analyses is broader internalizing symptoms (including depression, but also anxiety, withdrawal, somatic symptoms, etc). It might help to adjust the introduction and reduce the emphasis on depression specifically; or, if depression is the construct of interest, perhaps that subscale could be used instead of the broader internalizing subscale.

The integration of protective factors is important, but seems to be glossed over throughout the manuscript. There is only a brief sentence in the introduction, and no clear rationale for why certain protective factors were included. Because the authors state that they include protective factors in order to inform intervention policies and practices, I think this area warrants more attention.

Relatedly, there are a large number of analyses examining different facets of protective factors, including a factor score, then broken down into four domains, then using single items. Did the authors have hypotheses about these different approaches? How was the factor score constructed? Finally, some discussion regarding why some domains were protective but others were not seems warranted.

These analytic decisions seem to result in a very large number of models that were tested—did the authors consider correcting for multiple comparisons?

Can the authors provide brief rationale for the selection of covariates? E.g., are there race-based differences in the coupling between WHtR and internalizing symptoms that needed to be accounted for?

Page 5 states, "Finally, while BMI had a non-linear association (roughly U-shaped centered on a BMI~19, Supplementary Figure S1), WHtR was linearly associated with internalizing problems (Figure 1c), supporting its use as a practical transdiagnostic clinical measure." It seems reasonable to compare BMI and WHtR, but it's not clear to me how this result

indicates that WHtR is a more “practical transdiagnostic clinical measure”. Clarification would be helpful.

Reviewer #2 (Remarks to the Author):

I appreciate the opportunity to review this manuscript that investigates how early life environment, both adverse and protective experiences, affect body composition and internalizing problems coupling during adolescence. In this study, the authors found the waist-to-height ratio (WHtR) to be a stronger transdiagnostic measure than BMI in association with internalizing problems. The authors also found that early life adversity was significantly linked to WHtR-internalizing coupling, and that family and community protective experiences can buffer this coupling. Overall, this study addresses interesting research questions that have important clinical implications for physical and mental health. The authors provided sufficient information to document the study design and clear rationales for analytical decisions made in the study, and the manuscript is very well written. I am especially impressed by the authors' thorough considerations of alternative explanations and conducting a series of sensitivity analyses to determine the best analytical strategy. The level of detail provided in the manuscript and supplemental materials is sufficient for replication. Below, I offer a couple of comments, hoping to clarify some information and further strengthen this paper.

Introduction

- The introduction section provides a great background for this study. The only thing missing here is some discussion on potential protective factors. Right now, protective factors are mentioned very briefly in one sentence on page 4 (lines 104-106). Perhaps the authors can include some literature on potential protective factors in different contexts (family, peer, school, community – as tested in the study) and existing evidence on their associations with adolescent body composition & internalizing problems.
- Page 3, line 113-114, “we test the hypothesis that WHtR is a stronger predictor of adolescent mental health problems.” Please clarify stronger than what – I think the authors referred to BMI?

Methods & Supplemental Materials

- Supplementary Material S1: Data Preprocessing Procedures: More information is needed to describe the measures. For instance, citations of established measures such as CBCL, information on whether measures have been previously validated, and their reliability, internal consistencies, etc.
- There are some discrepancies when referring to tables, figures, or supplementary materials that need to be corrected:
 - o In the main text, Figure 1 was not cited. When referring to different panels of Figure 1, I believe the authors were actually talking about Figure 2.
 - o In “Supplementary Materials S2” section: “Model performance and additional explained variance over the parsimonious model were considered in aggregated (Table 2) and cross-sectionally (Figure 2d, Supplementary Figure S2). There is no Table 2 in the main text, so I am not sure which table the author was referring to. In addition, Figure 2d in the main text is about sibling discordance – was this cited correctly here?
 - o Supplementary Figure S2. “With the exception of the year 3 follow-up visit,” does this mean the 2y mark on the X axis, since the solid line here indicates that BIC (BMI + WHtR) < BIC (WHtR)?
 - o Protective environment general factor was S5 in the supplementary materials document but was cited as S2 in the main draft. Item-level moderation was S6 in the supplementary materials but cited as S3 in the main draft.
- Page 2 in supplementary materials - “Observations where participants chose not to respond were removed for analysis” – were they removed as long as they had missing data on one study variable? Citing Figure 1 in the main text here would be helpful (relatedly, the authors may also consider moving Figure 1 to the supplemental materials).
- Supplementary Figure S3: In panel B, please consider providing some notes explaining F (BMI|WHtR) and F (WHtR|BMI). I assume these F scores indicate model performance of one measure conditional on the other, but some explanation on how to interpret these scores and read the figure would be helpful.
- More information on the general protective environment factor (Supplementary Materials S5) would be helpful: was the one-factor structure derived from CFA or EFA? Please provide a measurement model and its fit indices.
- Supplementary Materials S6: After Bonferroni correction, shouldn't the p-value threshold be 0.0006 (0.05/82 comparisons)?
- Supplementary Figure S5, although the item labels provided in the table are helpful, it would be nice if the authors could label the figure items with their contents rather than variable names (e.g., fes_22_p relabeled as “Hard to blow off steam at home without upsetting somebody”)

Results

- Page 5, line 157 - “ACEs” had a large effect on WHtR-internalizing problems coupling. What criteria did the authors use to determine the magnitude (small/medium/large) of effects?
- Other than testing as moderators on the impact of ACEs, I am curious if the authors considered testing positive experiences in family, school, peer, and community contexts as direct predictors of body composition – internalizing problems coupling?

Discussion

- Page 6 lines 231-232 – “since many ACEs originate within the family, addressing these factors is vital but challenging.” I agree that community and peer-level interventions are equally important but think this statement could be addressed with more nuance. In particular, family experiences Aof CEs are not mutually exclusive with family supporting protective factors. It depends on many factors such as what type of ACEs experienced (child having critical illness but parents are supportive), timing (adversity in one period of childhood while supportive family in another developmental stage), who ACEs came from (substance abuse from one parent but support from the other parent). It is possible to buffer family-induced ACEs impacts using family-level intervention strategies.

* TRANSPARENT PEER REVIEW: Communications Psychology uses a transparent peer review system. This means that we publish the editorial decision letters including Reviewers' comments to the authors and the author rebuttal letters online as a supplementary peer review file. However, on author request, confidential information and data can be removed from the published reviewer reports and rebuttal letters prior to publication. If your manuscript has been previously reviewed at another journal, those Reviewers' comments would not form part of the published peer review file.

If you experience problems in linking your ORCID, please contact the Platform Support Helpdesk.

Version 1:

Decision Letter:

Dear Dr Rasmussen,

Your manuscript titled "Body Composition and Internalizing Problems in Adolescence: Moderation by the Early Life Environment" has now been seen by our reviewers, whose comments appear below. In light of their advice I am delighted to say that we are happy, in principle, to publish a suitably revised version in Communications Psychology.

We therefore invite you to revise your paper one last time to address the remaining concerns of our reviewers and a list of editorial requests. At the same time we ask that you edit your manuscript to comply with our format requirements and to maximise the accessibility and therefore the impact of your work.

EDITORIAL REQUESTS:

SUBMISSION INFORMATION:

OPEN ACCESS:

* TRANSPARENT PEER REVIEW: Communications Psychology uses a transparent peer review system. On author request, confidential information and data can be removed from the published reviewer reports and rebuttal letters prior to

publication. If you are concerned about the release of confidential data, please let us know specifically what information you would like to have removed. Please note that we cannot incorporate redactions for any other reasons.

* **DATA AVAILABILITY:**

Link Redacted

Best regards,

Jennifer Bellingtier

Jennifer Bellingtier, PhD
Senior Editor
Communications Psychology

REVIEWERS' EXPERTISE:

Reviewer #1 adversity, mental health, development
Reviewer #2 adversity, resilience, development

REVIEWERS' COMMENTS:

Reviewer #1 (Remarks to the Author):

Thank you to the authors for their thorough revisions and improvements to the manuscript. My comments have been addressed, but the additional detail regarding the protective factors raised a couple of other questions:

Apologies if I missed this detail, but for the family environment items, were both parent and child report included? Were the item-level child responses not significant? This would be worth explicitly noting.

The EFA results suggested that that a single factor is not fully explanatory, and the CFA did not have good model fit (RMSEA=0.113 [90% CI 0.100–0.128]; CFI=0.778...). Why then was this general factor included in the analyses? In the main text, it is reported as a significant moderator without this important caveat. This should be addressed.

Reviewer #2 (Remarks to the Author):

The authors have thoroughly addressed all my concerns. I really appreciate the opportunity to review this excellent paper!

Communications Psychology

MS Number: COMMSPSYCHOL-25-0365-T

Title: Body Composition and Internalizing Problems in Adolescence: Moderation by the Early Life Environment

We thank the editor and reviewers for their time spent expertly and insightfully reading the material to provide constructive feedback. In response, we have carefully prepared a revision of the original submission based on the reviewer's comments. In sum, we submit that this process has substantially improved the strength of evidence provided in the manuscript. Manuscript edits are denoted by **bold** text below.

Reviewer #1:

Major Comment R1.1a

"Throughout the manuscript, it would be helpful to have greater specificity in the description and labeling of constructs. For example, in the adversity literature, "ACEs" often refers to a specific list of exposures from the original CDC ACEs study. Because the metric of adversity in the present study is more comprehensive, it may be helpful to instead label the construct as early-life adversity (ELA)."

Response:

We thank the reviewer for this suggestion. We have replaced "ACEs" with "ABCD-ACEs" when used to describe the derived scores throughout the manuscript to clarify the approach used (Manuscript Ref 25) to align scores with the original ACEs study, but based on a more comprehensive assessment using ABCD Study data. Further, we have replaced "ACEs" with "ELA" when used conceptually, as appropriate. Finally, we now clearly state the qualifications of this verbiage for the reader in the introduction, and provide increased detail in the alignment of ABCD-ACEs with the standard ACEs questionnaire in the supplement.

[Manuscript Ref 25] Stinson EA, Sullivan RM, Peteet BJ, Tapert SF, Baker FC, Breslin FJ, et al. Longitudinal Impact of Childhood Adversity on Early Adolescent Mental Health During the COVID-19 Pandemic in the ABCD Study Cohort: Does Race or Ethnicity Moderate Findings? *Biol Psychiatry Glob Open Sci.* 2021;1(4):324–35.

Manuscript (Introduction) Page 5, Line 137. Edits in **bold**.

'Here, ELA exposure is operationalized using a previously described proxy (25) for traditional ACEs (26) measures based on a comprehensive set of ABCD study data ("ABCD-ACEs", see Supplementary Materials S1 for further details).'

Manuscript Throughout. Example use of ABCD-ACEs. Edits in **bold**.

"ABCD-ACEs had a significant effect on WHtR-internalizing problems coupling ($r^2=4.6\%$, or a ~ 0.1 s.d. increase in slope for each ABCD-ACE reported, Figure 3)."

Manuscript (Supplementary Materials S1: Moderators) Page 1. Edits in **bold**.

"This "ABCD-ACEs" score includes domains conceptually aligned with the original ACEs categories. However, the ABCD-ACEs score expands this framework to include additional adversities relevant to the ABCD cohort, such as exposure to discrimination, bullying, natural disasters, and significant accidents. These additions reflect a broader but still

conceptually grounded operationalization of childhood adversity using item-level data.'

Major Comment R1.1b

"Similarly, the introduction is largely focused on depression specifically, but the metric used in analyses is broader internalizing symptoms (including depression, but also anxiety, withdrawal, somatic symptoms, etc). It might help to adjust the introduction and reduce the emphasis on depression specifically; or, if depression is the construct of interest, perhaps that subscale could be used instead of the broader internalizing subscale."

Response:

The reviewer brings up an important point. We continue to believe that internalizing symptoms are an appropriate variable for consideration based on their clinical relevance as indicators of future mental health problems and their established associations with BMI. However, we agree that a reduced emphasis on depression is warranted along with a more expansive justification for the focus on internalizing symptoms. The introduction has been revised accordingly.

Manuscript (Introduction) Page 3, Line 67. Edits in **bold**.

*"Early risk factors—such as elevated body-mass-index (BMI) and waist-to-height ratio (WHtR) for metabolic disorders, and internalizing symptoms (**e.g., depression, anxiety, withdrawal**) for ~~depressive disorders~~ **mental health problems**—often emerge in childhood and adolescence(1)."*

Manuscript (Introduction) Page 3, Line 73. Edits in **bold**.

*"For example, one recent meta-analysis estimated that individuals with metabolic syndrome have a 49% higher likelihood of developing depression (**a pattern that may reflect a broader vulnerability to internalizing symptoms**), while those with depression have a 52% increased risk of developing metabolic syndrome(7)."*

Manuscript (Introduction) Page 5, Line 129. Edits in **bold**.

*"We focus on internalizing problems due to their clinical relevance as indicators of future **emotional distress and risk for later mental health disorders, including but not limited to depression**, and their established associations with BMI."*

Major Comment R1.2

"The integration of protective factors is important, but seems to be glossed over throughout the manuscript. There is only a brief sentence in the introduction, and no clear rationale for why certain protective factors were included. Because the authors state that they include protective factors in order to inform intervention policies and practices, I think this area warrants more attention."

Response:

We agree with the reviewer(s) (see R2.1) that protective factors should be discussed more thoroughly in the introduction. We have revised the introduction in this regard.

Manuscript (Introduction) Page 4, Line 109. Edits in **bold**.

*"**In this study, we operationalize protective factors based on a theoretical framework previously described by the ABCD Culture and Environment Workgroup(17), which identifies four telescoping domains of environmental influence (family, peers, school, and community) as central to shaping adolescent development. Youth who report higher family support tend to show lower levels of internalizing symptoms(18), including depression and anxiety, and adolescents who regularly share family meals have reduced odds of developing overweight or obesity over time(19). Similarly, greater peer support is associated with lower internalizing symptoms(18), and peer relationships have also been linked to reductions in BMI, as evidenced by recent meta-analytic findings(20). Within school contexts, stronger school connectedness has been shown to predict lower***

depressive and anxiety symptoms(21), and students with higher connectedness scores are more likely to meet physical activity recommendations(22). At the community level, adolescents who feel a stronger sense of neighborhood connection tend to report fewer depressive symptoms(23), and environments that promote community cohesion, such as those with walkable infrastructure and access to green space, are negatively associated with adolescent BMI(24). Thus, these four domains are well-suited for identifying modifiable protective experiences that may speak to variability in outcomes (*i.e.*, resilience) by buffering the effects of early adversity and inform targeted intervention strategies.”

Major Comment R1.3

“Relatedly, there are a large number of analyses examining different facets of protective factors, including a factor score, then broken down into four domains, then using single items. Did the authors have hypotheses about these different approaches? How was the factor score constructed? Finally, some discussion regarding why some domains were protective but others were not seems warranted.”

Response:

We thank the reviewer for this point and agree that an increased description of how these analyses were approached is needed. In brief, we hypothesized that these factors, in general, would be protective against the observed effects of adversity. We began by drawing from the literature pertaining to, and data elements available in, the ABCD Study. To test our general *a priori* hypothesis we performed an Exploratory Factor Analysis (now with Confirmational Factor Analysis fit indices, *see R2.7*) of the domain level scores to achieve a general factor score. In the presence of a positive finding, and towards providing insights that may help inform at the policy level, we proceeded to conduct follow-up analyses at the domain and individual item level. In response, we have revised the manuscript to add a brief description of this process, and, now include discussion regarding why some domains were protective but others were not.

Manuscript (Introduction) Page 5, Line 140. Edits in **bold**.

“Finally, we explore protective factors in this framework. **We do so by testing the hypothesis that protective factors, in general, buffer the association between ABCD-ACEs and the within-person relationship between body composition and internalizing problems. We then conduct follow-up analyses at the domain- and item-levels aimed at mitigating the negative consequences of early-life adversity.**”

Manuscript (Discussion) Page 11, Line 340. Edits in **bold**.

“In addition, targeting factors on the community and peer-level in interventions is equally important, **particularly as community- and school-based factors emerged as significant moderators. Notably, community-based factors demonstrated the largest effects both independently and when accounting for other domains. In contrast, school-based factors did not reach statistical significance. This may reflect limitations in measurement rather than the absence of a meaningful effect, suggesting that more targeted studies with refined school-level assessments are needed to evaluate their true impact.**”

Major Comment R1.4

“These analytic decisions seem to result in a very large number of models that were tested—did the authors consider correcting for multiple comparisons?”

Response:

We thank the reviewer for this point. Multiple comparisons were considered. Specifically, we performed families of analyses and the number of comparisons was considered within family, as appropriate. For example, analyses comparing model fits between BMI and WHtR were descriptive and do not require multiple comparisons correction. Similarly, analyses demonstrating associations between WHtR - internalizing symptoms coupling and adversity were either *a priori* parsimonious models or attempting to refute the findings through sensitivity analyses.

Directly related to reviewer concern, we provided follow-up analyses of positive factors. We began by identifying a single general factor (one comparison) to test our hypothesis about buffering by positive factors. Based on the observation that this general factor moderates the effect of ACEs, and that variance explained by the general factor was modest, we expanded analyses to consider effects at the domain level. Because follow-up analyses are partially descriptive in nature we report both uncorrected and corrected (Bonferroni, four comparisons) significance thresholds. Similarly, these analyses were then extended to item level analyses where we also report both uncorrected and corrected (Bonferroni, 82 comparisons, *see also R2.8*) significant effects. Finally, we note that because domain and item scores share variance across subjects, it is likely that Bonferroni adjustment as used here is overly conservative, further justifying the reporting (but not interpreting) of uncorrected effects alongside effects properly corrected for multiple comparisons.

Major Comment R1.5

“Can the authors provide brief rationale for the selection of covariates? E.g., are there race-based differences in the coupling between WHtR and internalizing symptoms that needed to be accounted for?”

Response:

Covariates were selected from prior literature based on the effects of socioeconomic status in the context of BMI and cognition to help provide interpretability. We have revised the manuscript for added clarity. In addition, we now supply the full model in the Supplement quantifying the strength of associations between the coupling and covariate terms (including a significant effect of race, see below).

Supplementary Materials S1: Data Preprocessing: Covariates. Edits in **bold**.

“These measures were selected **based on published literature examining socioeconomic status in the context of obesity and cognition (5)**, and included age at interview, biological sex at birth, puberty status (visit-level specific, ABCD provided categorical measure, pds_p_ss_female_category_2 and pds_p_ss_male_category_2), self-reported race (collapsed into Black, White, Other based on prior ABCD practices)(3) and ethnicity, area deprivation index (baseline reported), household income, and maximum parental education (values below high-school level were collapsed into a single category).”

Supplementary Materials S5. Edits in **bold**.

“Supplementary Materials S5: Full Model Results for Testing the Association Between ABCD-ACEs and Random Slopes (Within-Individual Coupling Between Waist-to-Height Ratio and Internalizing Symptoms)

The fully adjusted model reflecting the association between ABCD-ACEs and the within-individual coupling between WHtR and internalizing symptoms is provided below.

Independent Variable	Estimate	Std. Error	t-value
(Intercept)	-2.276	0.408	-5.573
ABCD-ACEs	0.639	0.024	26.597
Biological Sex at Birth (Male)	-0.073	0.094	-0.781
Area Deprivation Index	-0.003	0.003	0.193
Parental Education (ref. Bachelor's Degree)			
≤ High School Graduate	0.082	0.269	0.307
High School Graduate	-0.175	0.228	-0.768
GED or Equivalent	-0.386	0.312	-1.237
Some College	-0.488	0.168	-2.905
Associates Degree Occupational	-0.219	0.196	-1.119
Associates Degree Academic	-0.514	0.214	-2.394
Master's Degree	0.146	0.136	1.079
M.D. or Equivalent	0.291	0.245	1.188
Ph.D. or Equivalent	0.238	0.231	1.030
Household Income	0.080	0.030	2.709
Self-report Ethnicity (Hispanic)	-0.073	0.129	-0.562
Self-report Race (ref. Black)			
Other/Unknown	0.337	0.214	1.570
White	0.465	0.146	3.182

Supplementary Table S10. ABCD-ACEs and the Within-individual Coupling Between Internalizing Behaviors and Waist-to-Height Ratio (WHtR). ABCD-ACEs were associated with the within-individual coupling between WHtR and internalizing symptoms after adjustment for potentially confounding factors.”

Major Comment R1.6

Page 5 states, “Finally, while BMI had a non-linear association (roughly U-shaped centered on a BMI~19, Supplementary Figure S1), WHtR was linearly associated with internalizing problems (Figure 1c), supporting its use as a practical transdiagnostic clinical measure.” It seems reasonable to compare BMI and WHtR, but it’s not clear to me how this result indicates that WHtR is a more “practical transdiagnostic clinical measure”. Clarification would be helpful.”

Response:

We appreciate the reviewer’s note about clarification. In response, we have revised the manuscript in this regard.

Manuscript (Results), Page 8, Line 242. Edits in **bold**.

“Finally, while BMI had a non-linear association (roughly U-shaped centered on a BMI~19, Supplementary Figure S1), WHtR was linearly associated with internalizing problems (Figure 1c), supporting its use as a practical transdiagnostic clinical measure **by offering a more intuitive, monotonically increasing association.**”

Reviewer #2:

Major Comment R2.1

“The introduction section provides a great background for this study. The only thing missing here is some discussion on potential protective factors. Right now, protective factors are mentioned very briefly in one sentence on page 4 (lines 104-106). Perhaps the authors can include some literature on potential protective factors in different contexts (family, peer, school, community – as tested in the study) and existing evidence on their associations with adolescent body composition & internalizing problems.”

Response:

We agree with the reviewer(s) (see R1.2) that protective factors should be discussed more thoroughly in the introduction. We have revised the introduction in this regard.

Manuscript (Introduction) Page 4, Line 109. Edits in **bold**.

“In this study, we operationalize protective factors based on a theoretical framework previously described by the ABCD Culture and Environment Workgroup(17), which identifies four telescoping domains of environmental influence (family, peers, school, and community) as central to shaping adolescent development. Youth who report higher family support tend to show lower levels of internalizing symptoms(18), including depression and anxiety, and adolescents who regularly share family meals have reduced odds of developing overweight or obesity over time(19). Similarly, greater peer support is associated with lower internalizing symptoms(18), and peer relationships have also been linked to reductions in BMI, as evidenced by recent meta-analytic findings(20). Within school contexts, stronger school connectedness has been shown to predict lower depressive and anxiety symptoms(21), and students with higher connectedness scores are more likely to meet physical activity recommendations(22). At the community level, adolescents who feel a stronger sense of neighborhood connection tend to report fewer depressive symptoms(23), and environments that promote community cohesion, such as those with walkable infrastructure and access to green space, are negatively associated with adolescent BMI(24). Thus, these four domains are well-suited for identifying modifiable protective experiences that may speak to variability in outcomes (i.e., resilience) by buffering the effects of early adversity and inform targeted intervention strategies.”

Major Comment R2.2

“Page 3, line 113-114, ‘we test the hypothesis that WHtR is a stronger predictor of adolescent mental health problems.’ Please clarify stronger than what – I think the authors referred to BMI?”

Response:

The reviewer is correct, this sentence has been revised for additional clarity.

Manuscript (Introduction) Page 5, Line 131. Edits in **bold**.

“Based on the logic that central adiposity is more directly related to the metabolic and immune consequences of obesity than total adiposity or weight-based measures of body composition (i.e., BMI) alone(9), we test the hypothesis that WHtR (a normalized measure of central adiposity) is a stronger predictor of adolescent mental health problems **than BMI.”**

Major Comment R2.3

“Supplementary Material S1: Data Preprocessing Procedures: More information is needed to describe the measures. For instance, citations of established measures such as CBCL, information on whether measures have been previously validated, and their reliability, internal consistencies, etc.”

Response:

We appreciate the opportunity to expand on the description and operationalization of the measures used in the current analysis. In short, the CBCL is a routinely used research and clinical tool with high test–retest reliability, internal consistency, criterion validity and shows good agreement between maternal and paternal ratings. The manuscript has been revised to expand on the description of these measures.

Supplementary Materials S1: Data Preprocessing: Covariates. Edits in **bold**.

“The CBCL is routinely used for research and clinical purposes with intraclass correlations above .90 for interparent agreement, 1-week test-retest reliability, and inter-interviewer reliability(1,2).”

Major Comment R2.4

“There are some discrepancies when referring to tables, figures, or supplementary materials that need to be corrected”

Response:

We thank the reviewer for highlighting the regrettable figure/table referencing errors throughout. The necessary edits have been made and can be seen throughout tracked changes.

Major Comment R2.5

“Page 2 in supplementary materials - “Observations where participants chose not to respond were removed for analysis” – were they removed as long as they had missing data on one study variable? Citing Figure 1 in the main text here would be helpful (relatedly, the authors may also consider moving Figure 1 to the supplemental materials).”

Response:

The reviewer is correct, missing data was means for exclusion in the current study and reflected in Figure 2 (formerly Figure 1 due to reordering). We now cite Figure 2 in this passage.

Supplementary Materials S1: Data Preprocessing: Covariates. Edits in **bold**.

“Observations where participants chose not to respond were removed for analysis as shown in Figure 2.”

Major Comment R2.6

“Supplemental Figure S3: In panel B, please consider providing some notes explaining $F(BMI|WHtR)$ and $F(WHtR|BMI)$. I assume these F scores indicate model performance of one measure conditional on the other, but some explanation on how to interpret these scores and read the figure would be helpful.”

Response:

We thank the reviewer for this point and have revised the manuscript for clarity.

Manuscript Supplemental Figure S3 Caption. Edits in **bold**.

“F-statistics shown are from nested linear models, testing each anthropometric measure’s independent effect when the other is included as a covariate. $F(BMI | WHtR)$: F-statistic for BMI in a model adjusting for waist-to-height ratio. $F(WHtR | BMI)$: F-statistic for waist-to-height ratio in a model adjusting for BMI. Larger F values indicate a stronger unique association with the outcome.”

Major Comment R2.7

“More information on the general protective environment factor (Supplementary Materials S5) would be helpful: was the one-factor structure derived from CFA or EFA? Please provide a measurement model and its fit indices.”

Response:

We appreciate the opportunity to clarify this aspect of our analysis. The one-factor structure for the general protective environment factor was derived using an Exploratory Factor Analysis (EFA), specifically the *factanal* function in R, which utilizes a maximum likelihood approach. The measurement model derived from the EFA is presented through the factor loadings for each domain onto the single general factor, and its fit indices described through chi-squared statistics. For greater clarity, we now repeat analyses using a Confirmational Factor Analysis (CFA) to provide additional model fits. In response, we have revised the manuscript with these expanded details to provide the reader with a deeper understanding of how the general protective environment factor was constructed.

Supplementary Materials S6: Protective Environments General Factor. Edits in **bold**.

“A general factor score composed of the four protective environment domains (family, peers, school, community) was derived using an **Exploratory Factor Analysis (EFA)**. The one-factor (general) was specified **in R (factanal)** using a maximum likelihood estimate and the goodness-of-fit assessed using the chi-square test. **A second Confirmatory Factor Analysis (CFA) was conducted (lavaan) to provide additional model fits and indices. CFA fit was modest ($\chi^2(2)=185.6, p<10^{-10}$; RMSEA=0.113 [90% CI 0.100–0.128]; CFI=0.778; TLI=0.333; SRMR=0.045).**

The **EFA-derived** general factor explained roughly 13.4% of the total variance with **chi-squared statistics ($\chi^2(2)=185.6, p<10^{-10}$)** suggesting that a single factor is not fully explanatory. This then motivates the inclusion of analyses at the domain-specific level. Factor loadings were comparable across domains ($\beta_{fam}=0.30, \beta_{peers}=0.47, \beta_{school}=0.34, \beta_{comm}=0.33$).”

Major Comment R2.8

“Supplementary Materials S6: After Bonferroni correction, shouldn't the p-value threshold be 0.0006 (0.05/82 comparisons)?”

Response:

We thank the reviewer for identifying this issue, their calculation is of course correct. The incorrect threshold (0.006) appears twice *as a typo*: once in the description and once in the figure label. All other uses of this threshold are code-based and have been confirmed to use the correct threshold ($\alpha=0.05/82=0.0006$, *see also github code repository*). The typos have been addressed in the current revision (see also figure in R2.9).

Supplementary Materials S7: Item-Level Moderation. Edits in **bold**.

“Single item analyses consisted of 82 comparisons and therefore our significance threshold was adjusted using a Bonferroni correction ($p_{thresh}=0.0006$).”

Major Comment R2.9

“Supplemental Figure S5, although the item labels provided in the table are helpful, it would be nice if the authors could label the figure items with their contents rather than variable names (e.g., fes_22_p relabeled as “Hard to blow off steam at home without upsetting somebody”)”

Response:

In response to the reviewer’s suggestion we have added labels to multiple comparison corrected significant items. Items below threshold are left unlabeled due to constraints on space and clarity. However, the figure is followed by a lookup table (Supplementary Table S11) for easy reference.

Supplementary Materials S7: Item-Level Moderation. Relabeled figure.

Major Comment R2.10

“Page 5, line 157 - “ACEs” had a large effect on WHtR-internalizing problems coupling. What criteria did the authors use to determine the magnitude (small/medium/large) of effects?”

Response:

We appreciate this point and have adjusted the language to remove this relatively arbitrary qualifier.

Manuscript (Results) Page 9, Line 262. Edits in **bold**.

“ABCD-ACEs had a **significant** large effect on WHtR-internalizing problems coupling ($r^2=4.6\%$, or a ~ 0.1 s.d. increase in slope for each ACE reported, Figure 3).”

Major Comment R2.11

“Other than testing as moderators on the impact of ACEs, I am curious if the authors considered testing positive experiences in family, school, peer, and community contexts as direct predictors of body composition – internalizing problems coupling?”

Response:

We did not directly test the effect of positive experiences as predictors of body composition – internalizing problems coupling. Our initial hypothesis was centered on the effect of adversity and

subsequent analyses considered potential buffering of this effect through positive experiences. While we continue to believe that this is an appropriate course of investigation, we also appreciate that the reviewer's question is a natural one. For internal review purposes we have run the requested model and observed that the general factor score for positive experiences is significantly and negatively associated with the body composition – internalizing problems coupling ($t=-11.6$, $r^2=1.8\%$, note that this effect size is roughly one third that of the ABCD-ACEs association). While interesting and consistent with the literature, we believe that a full treatment of this material is both necessary and outside the scope of the current work and has not been included in this revision. However, because we agree that this is an important point, we have added language promoting its future investigation to the discussion.

*Manuscript (Limitations) Page 12, Line 368. Edits in **bold**.*

“While this study focused on adversity and its potential buffering through positive experiences, future work should also examine whether positive experiences in family, school, peer, and community contexts may directly shape the coupling between body composition and internalizing problems.”

Major Comment R2.12

“Page 6 lines 231-232 – “since many ACEs originate within the family, addressing these factors is vital but challenging.” I agree that community and peer-level interventions are equally important but think this statement could be addressed with more nuance. In particular, family experiences of ACEs (sp.) are not mutually exclusive with family supporting protective factors. It depends on many factors such as what type of ACEs experienced (child having critical illness but parents are supportive), timing (adversity in one period of childhood while supportive family in another developmental stage), who ACEs came from (substance abuse from one parent but support from the other parent). It is possible to buffer family-induced ACEs impacts using family-level intervention strategies.”

Response:

We agree with the reviewer that there is nuance to this point and that it is possible to buffer family-induced ACEs impacts using family-level intervention strategies. In response, we have revised this discussion section to more transparently support intervention at the family level.

*Manuscript (Discussion), Page 11, Line 334. Edits in **bold**.*

“Identifying environmental conditions that foster resilience against early adversity is critical to informing policy. Towards this, the current study contributes to the field by elucidating conditions that may buffer the negative impact of ELA on the coupling between body composition and internalizing problems. Specifically, the family-based factors identified as having a protective effect can serve as intervention targets to promote acceptance, unconditional support, belonging, and predictability within families, potentially dampening the biological stress response to ELA (32) and, in turn, dampen the coupling between WHtR and internalizing problems. **Strengthening these family-based factors, especially in children for whom ACEs originate within the family as is commonly the case (33), should be a priority with individually tailored interventions building on those family-based resources available.**”

Communications Psychology

MS Number: COMMSPSYCHOL-25-0365A

Title (Revised): Early life environment moderates association of body composition and internalizing problems in adolescence

We thank the editor and reviewers again for their time spent expertly and insightfully reading the material to provide constructive feedback. In response, we have carefully prepared a revision of the originally revised submission based on the remaining reviewer comment and editorial table requests. The recommendations provided by the reviewers' and editors have made a substantial towards improving the strength and presentation of evidence provided in the manuscript. Manuscript edits are denoted by **bold** text below, responses to the editorial requests are embedded in the table, separately. Both are reflected in the track changes version of the revised manuscript.

Reviewer #2:

Comment R2.1

"Apologies if I missed this detail, but for the family environment items, were both parent and child report included? Were the item-level child responses not significant? This would be worth explicitly noting."

Response:

We agree that youth report would be an interesting measure to look at in this context. However, we focused on parent report as the consortium recommended cohesion subscale was only available in the parent report version. This is now explicitly noted.

Supplementary Materials. Edits in **bold**.

"Representative measures were pulled from four consortium-recommended domains focused on family cohesion (ce_p_fes/fes_p_ss_cohesion_sum_pr), peer involvement (ce_y_pbp/pbp_ss_prosocial_peers), school risk and protective factors (ce_y_srpf/srpf_y_ss_ses), and social cohesion at the neighborhood level (ce_p_comc/comc_ss_cohesion_p). **Family cohesion subscales were not available for the youth report version of the family environment scale.**"

Comment R2.2

"The EFA results suggested that that a single factor is not fully explanatory, and the CFA did not have good model fit (RMSEA=0.113 [90% CI 0.100–0.128]; CFI=0.778...). Why then was this general factor included in the analyses? In the main text, it is reported as a significant moderator without this important caveat. This should be addressed."

Response:

We agree that the CFA did not have a good model fit. However, we still believe that a single summary score is pragmatically supportive of positive environments, in general, in this context as a moderator consistent with the domain and item level models. In response, we now make this important caveat more explicit throughout by mentioning the poor fit and replacing reference of a "general factor" with "summary score". Internally, we note that this summary index produced by

the latent factor analysis is highly consistent with a composite index produced by Principal Components Analysis (summary score shares 97% variance with the first Principal Component).

*Example Edit. Manuscript (Methods) Page 9. Edits in **bold**.*

“These models were first constructed based on a Factor Analysis **summary score intended to be reflective of the overall quality of protective environments** (see *Supplementary Materials S6: Protective Environment **Summary Score***). Building on findings from the **summary** score analysis suggesting **that a single score is not fully explanatory of such positive environments**, we repeated the models to gain more fine-grained insight by examining scores at different social levels(17) (family, peers, school, community), as well as at the item level (see *Supplementary Materials S7: Item-Level Moderation of the Association Between ABCD-ACEs and Random Slopes*).“